# Translation and validation of the Chinese version of EORTC QLQ-SWB32 assessing the spiritual wellbeing of women with gynecological cancer

**Yue Feng**[1,2], **Jiangshan Luo**[3], **Tangwei Lin**[1,2], **Xingcan Liu**[1,2], **Xiujing Guo**[1,2], **Jing Chen**[1,2]*, **Bella Vivat**[4]

1 Department of Gynecological Nursing, West China Second University Hospital, Sichuan University, Chengdu, China, 2 Key Laboratory of Birth Defects and Related Diseases of Women and Children (Sichuan University), Ministry of Education, Chengdu, China, 3 Sichuan Provincial People's Hospital, University of Electronic Science and Technology of China, Chengdu, China, 4 Marie Curie Palliative Care Research Department, Division of Psychiatry, UCL, London, United Kingdom

* chenjing_scu@sina.com

## Abstract

### Background

This study aimed to translate the internationally developed and validated European Organization for Research and Treatment of Cancer measure of spiritual wellbeing (EORTC QLQ-SWB32) into Chinese, validate the translation with women with gynecological cancer, and examine associations between demographic variables and the scales of the measure.

### Methods

The study followed EORTC translation guidelines. After pilot testing with sixteen gynecological cancer patients, we validated the final measure with another 200 patients. We analyzed reliability using Cronbach's alpha coefficients. Exploratory factor analysis (EFA), confirmatory factor analysis (CFA), and exploratory graphic analysis (EGA) were used to analyze the construct validity. A multiple linear regression model analyzed the relationship of the factors to spiritual well-being.

### Results

Cronbach's alpha coefficients showed good reliability, ranging from 0.885 to 0.907 in each dimension. The EFA (KMO = 0.876, $\chi^2$ = 2865.036, df = 231, $P < 0.001$) and EGA produced a four-dimension structure. CFA fit statistics indicated adequate fit to a four-dimension solution ($\chi^2$/df = 2.178, RMESA = 0.077, GFI = 0.973, SRMR = 0.057, CFI = 0.915, TLI = 0.902), which matched the dimensions and constituent items from the original measure. Regression analysis indicated that higher education levels correlated with higher scores on the Relationships with Others (RO) and Existential (EX) scales; unemployment with lower

**Data availability statement:** All relevant data are within the paper and its Supporting Information files. The EORTC QLQ-SWB32 is free for academic use; however, potential users must sign a user agreement with the Group to obtain permission for its use. The agreement and the full content of the Chinese version can be requested through the website: https://qol.eortc.org/questionnaire/qlq-swb32/.

**Funding:** 1. Initials of the authors who received each award: Jing Chen 2. Grant numbers awarded to each author: 21PJ056 3. The full name of each funder: Health Commission of Sichuan Province, China 4. Did the sponsors or funders play any role in the study design, data collection and analysis, decision to publish, or preparation of the manuscript? No.

**Competing interests:** The authors have declared that no competing interests exist.

Relationship with Self (RS) scores, and lower incomes with lower EX scores; patients with religious beliefs scored higher on Relationship with God (RG).

## Conclusions

The Chinese EORTC QLQ-SWB32 exhibits good reliability and validity among gynecological cancer patients, with dimensions aligning with those found in the original validation. This approved, validated instrument is now available for Chinese medical staff to use to assess the spiritual wellbeing of Chinese cancer patients and help improve understanding of the relevance of spiritual wellbeing to people from Chinese cultural backgrounds.

## 1 Background

Gynecological cancers pose serious and potentially fatal risks. In 2020, around 250,000 women in China were newly diagnosed with gynecological cancer [1]. Among these, cervical, ovarian, and endometrial cancers were among the top ten cancers diagnosed in Chinese women [2]. As females with cancer have been reported to be more adversely affected than men [3], it's vital to help gynecological cancer patients improve their quality of life and mental health.

Spiritual wellbeing (SWB) is an essential dimension of health-related quality of life for cancer patients [4]. It has been defined in various ways, including 'meaning, wholeness, transcendence, connection, joy, and peace, that does not rely on one's participation in an organized religion' [5]. Studies report that SWB correlates with clinically relevant outcomes such as anxiety, depression, end-of-life coping, and physical health [6–8].

There has been increasing clinical and research interest in SWB in the past decades. Spirituality has been conceptualized as an aspect of quality-of-life and spiritual well-being instruments such as Functional Assessment of Chronic Illness Therapy-Spiritual Well-Being (FACIT-Sp) [9], and Mental, Physical, and Spiritual Well-Being Scale (MPS) [10] have been developed. However, a meta-analysis has argued that spirituality is best seen as a distinct, although related, concept from quality of life [11]. There is as yet no "gold standard" measure of SWB, and some of the most widely used measures have been developed and validated in single cultural contexts, or a religious framework [12–14], so translating and applying these instruments in different cultures may not be simple [15]. It has been argued that measures of spiritual wellbeing should be developed and validated cross-culturally, and in the languages in which they are likely to be used. This means that any linguistic and conceptual difficulties can be resolved during development, rather than in later field-testing and validation [16,17].

Members of the European Organization for Research and Treatment of Cancer (EORTC) Quality of Life Group recently completed international validation of a stand-alone measure of spiritual wellbeing for people receiving palliative care for cancer: the EORTC QLQ-SWB32 (SWB32) [17]. The SWB32 was developed based on relevant literature, expert opinions, and interviews with palliative cancer patients [12,16,17] and validated in a multilingual and cultural context involving 14 countries, including China [17]. It has four main scoring dimensions, or scales: Relationship with Self, Relationships with Others, Relationship with Someone or Something Greater, and Existential. It was designed to suit people with various religious faiths or spiritual beliefs and those with none [17].

Chinese culture traditionally lacks a formalized religious structure, but philosophies such as Buddhism, Taoism, and Confucianism have significantly influenced both spiritual development and secular culture in China. Consequently, the boundaries between Chinese philosophical beliefs, religions, notably Buddhism and Taoism, and local folk religious practices

blur considerably [18]. While many Chinese people claim no formal religious beliefs but, nevertheless, influenced by the context of traditional Chinese culture and philosophies [19], exhibit religion-like beliefs and behaviors [18]. A notable manifestation of this is evident in the narratives of numerous Chinese patients who invoke Karma to rationalize their hospitalization experiences, attributing their diseases to a predetermined fate orchestrated by higher cosmic forces at the moment of their birth [20]. Taoists include veneration of ancestors and local gods in seeking resolutions to life's challenges [21]. However, it is noteworthy that many Chinese individuals also pray to deities rooted in Taoism, aligning their petitions with specific desires. These desires encompass a diverse spectrum, from seeking the favor of Gods associated with wealth, health, academic success, environmental harmony, pregnancy, and culinary pursuits [21]. This multifaceted spiritual background confers a spiritual dimension akin to conventional religious systems upon the Chinese [22].

Chinese culture traditionally emphasizes harmony, collectivism, and moral ethics, which significantly influence spiritual perceptions [23]. Unlike Western conceptions of spirituality that often center on personal relationships with a transcendent deity, Chinese spirituality is deeply rooted in the concepts of interconnectedness and balance [23,24]. This is reflected in Confucianism's emphasis on social harmony, Taoism's pursuit of unity with nature, and Buddhism's focus on inner enlightenment. These philosophical traditions contribute to a holistic worldview where spirituality is integrated into everyday life rather than confined to organized religious practices [25]. Consequently, spirituality in Chinese culture is more likely to be expressed through ethical living, social roles, and relational harmony rather than explicit religious beliefs [25].

Researchers have recognized the importance of spirituality within Chinese contexts, defining it as a relationship with self, others, nature, and Higher Being(s) [26,27]. These dimensions align with the four key scoring areas of SWB32. The international validation study included Chinese participants, but at that time, the Chinese translation of the SWB32 had not yet been approved by the EORTC Translation Unit [28,29]. We conducted this later study to produce an approved Chinese translation, explore its reliability and validity with Chinese gynecological cancer patients, and explore associations between spiritual well-being and demographic characteristics for these patients. This approved, validated instrument is now available for Chinese medical staff to use to assess the spiritual wellbeing of Chinese cancer patients, and help improve understanding of the relevance of spiritual wellbeing to people from Chinese cultural backgrounds.

## 2 Methods

### 2.1 Translation procedure

The translation from English to Chinese was performed in collaboration with the EORTC Translation Unit, following the standard EORTC translation procedure [30]: two forward translations, initial linguistic reconciliation by our research group, two backward translations of the reconciled version, and then EORTC Translation Unit review of the translation report. Following several rounds of review, discussion, and revisions, our Chinese translation was approved for pilot testing.

We conducted two rounds of pilot testing with cancer patients in March 2023. Patients completed the questionnaire and were then interviewed face-to-face about their feelings and experiences to check their understanding. The questions were: "Do you understand these items? Is the item confusing? Are there any difficult words? Is the item upsetting? If the answer is yes, how would you ask this question?" We amended any problematic items as necessary and tested the new translation with a new group of patients. The final version,

approved by the EORTC Quality of Life group as the official Chinese version of the EORTC QLQ-SWB32, went forward to larger-scale validation.

## 2.2  Validation phase

For the validation study, we continuously recruited patients from April 1st, 2023, to August 31st, 2023, in a women's and children's medical center in western China. Patients were enrolled if they 1) had been diagnosed with gynecological cancer, such as ovarian, cervical, endometrial, and fallopian tube cancer; 2) could understand and answer relevant questions. Patients were excluded if they 1) refused to participate in this study or 2) had cognitive impairment. We collected relevant demographic information, including age (years), ethnicity, marital status, employment status, educational background, and monthly household income (Yuan). Clinical data, such as diagnosis and cancer stage, were extracted from patients' medical records. Additionally, we inquired whether patients held religious beliefs or actively engaged in religious practices. We helped patients fill in the questionnaires, along with an explanation of the concept of spiritual well-being. 'Spiritual' was described as a relation to people's thoughts, beliefs, faith, and connections between self and one's outlook on life. 'Well-being' refers to being in a good state. 'Spiritual wellbeing' means being content with oneself according to one's own beliefs. It could be finding meaning and purpose in life through a connection with oneself, other people, art, music, literature, nature, prayer, meditation, or a power greater than oneself; harmony between mind, body, and soul; the affirmation of life in a relationship with God, the self, the wider environment and a good balance between one's beliefs and one's actions.

This study was performed following the principles of the Declaration of Helsinki. The Ethics Committee of West China Second University Hospital of Sichuan University approved the study protocol (No. 2019-13). Written informed consent was obtained from all the patients included in this study.

## 2.3  EORTC QLQ-SWB32 and its scoring scales

The SWB32 has 32 items, with 22 of them comprising its four main scoring scales: Existential (6 items), Relationship with Self (5 items), Relationships with Others (6 items), and Relationship with Someone or Something Greater (5 items). These items are scored from 1 (not at all) to 4 (very much). Items 22 and 23 are non-scoring items that serve to identify respondents with a belief for whom the single-item scale Relationship with God (item 26) is applicable. Only these individuals should respond to that item. The remaining items are six further non-scoring but clinically relevant items, plus a single item for overall spiritual wellbeing, or Global SWB, which is scored from 1 (very poor) to 7 (excellent), with an additional choice of 0 (cannot reply or don't know) [14]. The Relationship with Self scale is reverse scored. Total scores of the multi-item scales are transformed into centesimal scores (the score/possible highest score*100). All scales then range from 0 to 100, with high scores representing positive outcomes.

## 2.4  Statistical analysis

We present our study data as frequencies (percentages) and means plus standard deviations. The distribution of included SWB32 scores was skewed, so we used the Spearman correlation coefficient to calculate correlations between scores on each SWB32 scale and patients' ages. We assessed relationships between spiritual wellbeing and countable variables using the Nonparametric Test. Mann-Whitney U tests were used for comparisons between two groups; Kruskal- Wallis H tests were used for more than two groups. We included all variables with

*P* < 0.1 in the correlation analysis in the multi-linear regression analysis to investigate possible factors associated with spiritual well-being. Statistical tests, all conducted using SPSS 27.0, were two-tailed, with statistical significance set at an alpha level of 0.05.

**2.4.1 Reliability.** We assessed the internal consistency of the SWB32 using Cronbach's alpha coefficients (recommended value > 0.7) and split-half reliability (recommended value > 0.8). We conducted item analysis using Spearman correlation coefficients to assess the correlations between each item and the total score of scales (recommended value > 0.4) [31].

**2.4.2 Content validity.** Six healthcare practitioners in palliative and cancer care assessed the questionnaire's clarity and semantics. They should have a master's degree or above, at least 5 years of experience in this research field, and good command of English. They provided suggestions for any modifications they thought necessary.

**2.4.3 Construct validity.** We conducted Exploratory Factor Analysis (EFA) to help determine the underlying theoretical structure of the scale, excluding the non-scoring items and the single Global SWB item. We applied Bartlett's test of Sphericity and Kaiser-Meyer-Olkin (KMO) measure of sampling adequacy before the EFA, to verify the appropriateness [32]. We used principal axis factoring and oblique rotation for all responses. After the EFA, we conducted a confirmatory factor analysis (CFA) to confirm the EFA-suggested factor structure. We used JASP 0.17.2.1 [33] for both the EFA and CFA. We evaluated goodness of fit using the following indices [34]: chi-squared value to degrees of freedom ratio ($\chi^2/df$) < 3, the Goodness of Fit Index (GFI) > 0.9, the root mean squared error of approximation (RMESA) < 0.08, the standardized root mean squared residual (SRMR) < 0.08, the Comparative Fit Index (CFI) > 0.9, and Tucker-Lewis Index (TLI) > 0.9.

**2.4.4 Exploratory graph analysis (EGA).** We applied EGA, using R software 4.3.1 and R packages EGAnet [35], to inspect SWB32 dimensionality. EGA estimates a Gaussian graphic model with LASSO and then uses the Walktrap algorithm [36]. It produces a visual network plot to indicate the number of dimensions of SWB32, which items cluster together, and their level of associations [35]. Each node in the network represents each SWB32 item, while edges present the partial correlation between two items. The magnitude of the correlation is represented by the thickness of the edges [37]. This plot can discover the dimensionality of the SWB32 by identifying the number of item clusters [38], and we investigated whether the clusters generated by the network were consistent with the dimensions of the English version of the measure.

## 2.5 Inclusivity in global research

Additional information regarding the ethical, cultural, and scientific considerations specific to inclusivity in global research is included in the Supporting Information.

## 3 Results

### 3.1 Participant characteristics

We conducted preliminary pilot testing with ten cancer patients: three with ovarian, four cervical, and three with endometrial cancers (mean age 50.80 ± 12.20 years, ranging from 31 to 70). Four patients reported difficulty understanding Q2, 'I have felt at peace with myself,' so we revised the Chinese translation following their comments and suggestions. This item can be understood as feeling free from emotional or mental agitation, a sense of calm or feeling untroubled, so it was described as '平和' in Chinese (back translation to English: calm and peace). The second round of pilot testing involved six other cancer patients (39 to 59 years old). These patients' responses indicated that they well understood the final translation. Four

female and two male healthcare professionals in palliative and cancer care (ranging from 30 to 47 years old) assessed content validity.

Our final validation involved 200 patients, with no missing data. Their mean age was 50, (SD 12). Most (85%) were married, and half (50%) were unemployed. 41.5% of the patients had cervical cancer, with 22.5% ovarian/fallopian tube cancer, and 28.5% endometrial cancer. Most (59%) had Stage I cancer, with 10.5% stage II, 13.0% stage III, and 5.5% stage IV. 96.5% of the patients declared that they had no religious beliefs and were not actively engaged in religious practices; one (0.5%) was Christian, and six (3.0%) were Buddhist. Table 1 presents patients' demographic characteristics.

### 3.2 Reliability and validity

**3.2.1 Reliability.** Table 2 shows the means and SDs for each item. Correlation coefficients between each item and the total score ranged from 0.726 to 0.883 ($P < 0.05$). The Cronbach's alpha values (RO 0.888, RS 0.885, RSG 0.907, EX 0.897) and Guttman split-half coefficients (RO 0.829, RS 0.839, RSG 0.890, EX 0.814) indicated good internal reliability.

**3.2.2 Content validity.** All healthcare professionals interviewed found the language easy to understand and the content appropriately relevant to palliative and cancer care.

**3.2.3 Construct validity.** The KMO value (0.876) and the Barlett Sphericity test results ($\chi^2 = 2865.036$, df $= 231$, $P < 0.001$) were suitable for factor analysis. Table 3 presents item loading results, indicating four factors of SWB32. The exploratory graph analysis (EGA) produced a four-dimensional structure of the SWB32 (Fig 1), showing the connectedness of the factors and item clusters as a network, with the width of the lines indicating the strength of the relationships between items. The EGA-identified dimensions and the items in each cluster were equivalent to the scales in the internationally validated SWB32. The fit statistics in the confirmatory factor analysis (CFA) indicated an adequate fit for a four-factor solution ($\chi^2$/df $= 2.178$, RMESA $= 0.077$, GFI $= 0.973$, SRMR $= 0.057$, CFI $= 0.914$, TLI $= 0.902$). The single Relationship with God item was not included in CFA as the model couldn't be estimated with one observed variable in one factor.

### 3.3 Associated factors of spiritual well-being

We analyzed the association between SWB32 scale scores and respondents' demographic characteristics (Table 4). Table 5 presents the results from the multiple linear regression analysis, including variables with $P < 0.1$ in the single-factor analysis seeking to identify factors associated with spiritual well-being. We found that high Relationships with Others scores were associated with college and higher education (B $= 6.860$, $P = 0.016$). Employed or retired respondents had higher Relationship with Self scores than unemployed respondents (B $= 5.594$, $P = 0.037$ v.s B $= 6.036$, $P = 0.040$). Higher Existential scores were associated with senior or college and above education (B $= 6.970$, $P = 0.027$ v.s B $= 8.668$, $P = 0.009$), family income more than 5000 per month (B $= -7.807$, $P = 0.023$) compared to junior and below, and income less than 3000. As for Relationship with God score, patients with religious beliefs (B $= 30.315$, $P < 0.001$), were less educated (B $= -10.969$, $P = 0.019$), and those living in urban areas had higher RS scores (B $= -10.221$, $P = 0.009$). The independent variables in this multiple regression analysis explained 4.9% of the variance in RO, 4.1% in RS, 10.1% in EX, and 15.6% in RG.

## 4 Discussion

We translated the EORTC QLQ-SWB32 to Chinese following EORTC translation guidelines, and our translation was approved by the EORTC Translation Unit. We then tested it on Chinese gynecological cancer patients, confirming internal reliability and validity. We identified

**Table 1. Patients' demographic and clinical characteristics (N = 200).**

| Variables | n (%)/ means (SD) |
|---|---|
| Age (years) | 50.16 ± 12.22 |
| Ethnic group | |
| Han | 187 (93.5) |
| Tibetan | 4 (2.0) |
| Others | 9 (4.5) |
| Religious belief | |
| None | 193 (96.5) |
| Buddhist | 6 (3.0) |
| Christian | 1 (0.5) |
| Marriage status | |
| Married | 170 (85.0) |
| Unmarried/divorced/widowed | 30 (15.0) |
| Education background | |
| Elementary school and below | 50 (25.0) |
| Junior high school | 73 (36.5) |
| Senior middle school | 35 (17.5) |
| College and above | 42 (21.0) |
| Work status | |
| Retired | 39 (19.5) |
| Unemployed | 101 (50.0) |
| Employed | 60 (30.0) |
| Current Caregiver | |
| Partner | 98 (49.0) |
| Family member | 102 (51.0) |
| Monthly household income (Yuan) | |
| <3000 | 45 (22.5) |
| 3001–5000 | 84 (42.0) |
| >5000 | 71 (35.5) |
| Residence | |
| Urban area | 133 (66.5) |
| Rural area | 67 (33.5) |
| Primary cancer | |
| Cervical cancer | 83 (41.5) |
| Ovarian/fallopian tube cancer | 45 (22.5) |
| Endometrial cancer | 57 (28.5) |
| Others | 15 (7.5) |
| Cancer stage | |
| I | 118 (59.0) |
| II | 21 (10.5) |
| III | 26 (13.0) |
| IV | 11 (5.5) |

some factors, including level of education, employment status, and religious beliefs, which are associated with our gynecological cancer patients' scores on the SWB32 scales.

The internal consistency and split-half reliability were generally strong for four factors, and the results were similar to the internationally validated version [17]. The four-factor structure

**Table 2. Correlation between items and the total scores of each EORTC QLQ-SWB32 scale.**

| Scales | Means (SD) | EX | RO | RS | RSG | RG | Global-SWB |
|---|---|---|---|---|---|---|---|
| **Existential (EX)** | 73.21 (15.69) | 1.000 | | | | −0.084 | 0.519** |
| Q1: able to deal with problems | 66.00 (19.74) | 0.759** | | | | | |
| Q2: peace with myself | 71.88 (18.91) | 0.795** | | | | | |
| Q3: find things I enjoy | 71.38 (19.48) | 0.805** | | | | | |
| Q14: my life is fulfilling | 79.13 (18.03) | 0.834** | | | | | |
| Q15: my life is worthwhile | 79.00 (18.32) | 0.840** | | | | | |
| Q16: plan for the future | 71.88 (21.26) | 0.834** | | | | | |
| **Relationships with others (RO)** | 79.75 (14.70) | 0.588** | 1.000 | 0.213** | 0.322** | 0.018 | 0.388** |
| Q8: share thoughts with those close to me | 78.38 (18.36) | | 0.726** | | | | |
| Q9: loved by those important to me | 84.25 (16.87) | | 0.824** | | | | |
| Q10: someone to talk to about my feelings | 78.88 (18.77) | | 0.818** | | | | |
| Q11: able to trust others | 78.25 (18.81) | | 0.794** | | | | |
| Q12: able to forgive others | 75.88 (19.15) | | 0.748** | | | | |
| Q13: valued as a person | 82.88 (18.17) | | 0.800** | | | | |
| **Relationship with someone/something greater (RSG)** | 56.93 (16.26) | 0.279** | | | 1.000 | 0.129 | 0.299** |
| Q20: time for quietness/prayer/meditation | 56.88 (17.89) | | | | 0.834** | | |
| Q21: important others pray for me | 57.25 (21.50) | | | | 0.872** | | |
| Q27: live on through words, deeds... | 58.50 (18.32) | | | | 0.786** | | |
| Q30: I believe in life after death | 47.88 (19.70) | | | | 0.883** | | |
| Q31: I have spiritual wellbeing | 64.13 (17.47) | | | | 0.845** | | |
| **Relationship with self (RS)** | 78.85 (14.73) | 0.338** | | 1.000 | 0.066 | 0.082 | 0.268** |
| Q5: troubled | 77.75 (16.59) | | | 0.742** | | | |
| Q6: lonely | 86.50 (17.87) | | | 0.829** | | | |
| Q17: worries/concerns about the future | 76.50 (18.01) | | | 0.805** | | | |
| Q18: can anything be done for me | 74.25 (16.23) | | | 0.777* | | | |
| Q19: unfair that I am ill | 79.25 (20.07) | | | 0.806** | | | |
| **Relationship with God (RG)** | 44.06 (19.30) | | | | | −0.105 | |
| **Global-SWB** | 68.14 (27.09) | | | | | | |

identified from EFA, CFA, and EGA in our study aligns closely with the international study's principal component and Rasch-derived scales: Relationships with Others (RO), Relationship with Self (RS), Relationship with Someone or Something Greater (RSG), and Existential (EX). EFA loaded all the items on the same factors as in the internationally validated version. For the Relationship with God (RG) single-item scale, the two non-scoring "skip" items that determine whether patients should respond to the RG item were not included in the EFA. The single RG item didn't load on any factor and was separate from the four main dimensions. The CFA and EGA both also confirmed the structure of four dimensions.

Two previous studies employing the Chinese translation that had not been approved by the EORTC Translation Unit, reported on levels of spiritual well-being among their cancer patients (all palliative patients) [28,29,39]. Sun's study [29,39] did not convert the total scores of the multi-item scales into centesimal scores, prompting us to compare the raw scores across these studies. The other study, involving members of our group [28], and also using the unapproved translation version, exhibited scores on each scale that were notably akin to the findings in our current study, both of which explored SWB with gynecological cancer patients. In contrast, Sun et al. [29,39] included both male and female patients with a diverse range of

**Table 3. Results of exploratory factor analysis using principal axis factoring and oblimin rotation.**

| SWB32 | Item | EX | RO | RSG | RS | Uniqueness |
|---|---|---|---|---|---|---|
| EX: able to deal with problems | 1 | 0.760 | | | | 0.497 |
| EX: peace with myself | 2 | 0.686 | | | | 0.453 |
| EX: find things I enjoy | 3 | 0.763 | | | | 0.396 |
| EX: my life is fulfilling | 14 | 0.718 | | | | 0.326 |
| EX: my life is worthwhile | 15 | 0.686 | | | | 0.320 |
| EX: plan for the future | 16 | 0.785 | | | | 0.373 |
| RO: share thoughts with those close to me | 8 | | 0.568 | | | 0.511 |
| RO: loved by those important to me | 9 | | 0.801 | | | 0.330 |
| RO: someone to talk to about my feelings | 10 | | 0.730 | | | 0.383 |
| RO: able to trust others | 11 | | 0.772 | | | 0.424 |
| RO: able to forgive others | 12 | | 0.683 | | | 0.518 |
| RO: valued as a person | 13 | | 0.770 | | | 0.336 |
| RSG: time for quietness/prayer/meditation | 20 | | | 0.772 | | 0.385 |
| RSG: important others pray for me | 21 | | | 0.840 | | 0.297 |
| RSG: live on through words, deeds... | 27 | | | 0.728 | | 0.428 |
| RSG: I believe in life after death | 30 | | | 0.912 | | 0.176 |
| RSG: I have spiritual well-being | 31 | | | 0.776 | | 0.325 |
| RS: troubled | 5 | | | | 0.671 | 0.457 |
| RS: lonely | 6 | | | | 0.851 | 0.223 |
| RS: worries/concerns about the future | 17 | | | | 0.797 | 0.378 |
| RS: can anything be done for me | 18 | | | | 0.812 | 0.372 |
| RS: unfair that I am ill | 19 | | | | 0.739 | 0.405 |
| RG: I feel connected to God or someone or something greater than myself | 26 | | | | | 0.954 |

Abbreviations: RO, relationships with others; EX, existential; RS, relationship with self; RSG, relationship with someone/something greater; RG, relationship with God; SWB, spiritual well-being.

Note: The RG scale was answered only by the patients who responded 2–4 to items 21 and 22 (n = 122).

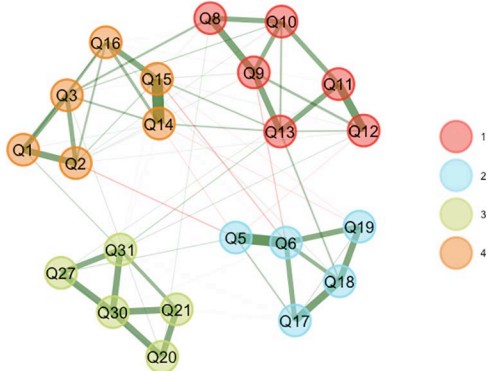

**Fig 1. Exploratory graph analysis of the four factors of SWB32.** † Colored cluster: Red (group 1), Relationships with Others; Blue (group 2), Relationship with Self; Green (group 3), Relationship with Someone or Something Greater; Orange (Group 4), Existential. ‡ Nodes (circles) represent the items in SWB32. Each node corresponds to a specific item measuring different aspects of spiritual wellbeing. Edges (lines) represent partial correlations between items. The thickness of the edges represents the magnitude of the correlation; the thicker the edge, the stronger the correlation. Green edges = positive correlations, red edges = negative correlations. § The RG scale was answered only by the patients who responded 2-4 on items 21 and 22 (n = 122).

**Table 4. Correlation between SWB32 scales and patients' characteristics[†].**

| | | RO | RS | RSG | EX | RG‡ | | Global SWB |
|---|---|---|---|---|---|---|---|---|
| | | | | | | n (%) | mean (SD)/r | |
| Age (Years) *continuous* | | -0.066 | **0.150*** | 0.093 | -0.044 | **0.218*** | | -0.101 |
| Ethnic group | | -0.882 | -0.502 | -0.230 | -0.547 | -0.521 | | -0.775 |
| | Han | 80.10 (14.22) | 78.93 (14.87) | 56.87 (16.05) | 73.33 (15.74) | 113 (92.6) | 43.58 (18.53) | 68.45 (27.13) |
| | Minority | 74.68 (20.52) | 77.69 (13.01) | 57.69 (19.75) | 71.47 (15.58) | 9 (7.4) | 50.00 (27.95) | 63.74 (27.11) |
| Religious belief | | -0.244 | -0.060 | -1.197 | -0.874 | **-2.618*** | | -0.599 |
| | No | 79.66 (14.73) | 79.02 (14.33) | 56.58 (16.12) | 73.36 (15.60) | 116 (95.1) | 42.67 (17.76) | 68.39 (26.89) |
| | Yes | 82.14 (14.97) | 74.29 (24.57) | 66.43 (18.42) | 69.05 (18.92) | 6 (4.9) | 70.83 (29.23) | 61.22 (33.72) |
| Marriage status | | -0.875 | -0.103 | -0.461 | -1.375 | -0.436 | | **-1.693** |
| | Married | 80.10 (14.85) | 78.74 (14.96) | 57.03 (16.29) | 73.85 (15.91) | 107 (87.7) | 43.46 (18.28) | 69.58 (26.26) |
| | Others | 77.78 (13.94) | 79.50 (13.60) | 56.33 (16.34) | 69.58 (14.10) | 15 (12.3) | 48.33 (25.82) | 60.00 (30.57) |
| Education background | | **10.984*** | 3.838 | 2.832 | **19.179**** | **6.140*** | | 3.082 |
| | Junior and below | 77.03 (15.28) | 77.97 (16.02) | 57.48 (15.88) | 69.44 (15.86) | 74 (60.7) | 46.62 (18.19) | 65.16 (28.87) |
| | Senior middle school | 82.86 (14.50) | 82.71 (13.08) | 53.29 (15.76) | 77.86 (14.67) | 22 (18.0) | 37.50 (14.94) | 75.51 (20.37) |
| | College and above | 85.12 (10.97) | 78.21 (11.52) | 58.33 (17.66) | 80.36 (12.46) | 26 (21.3) | 42.31 (24.26) | 70.75 (25.62) |
| Work status | | 5.516 | **6.639*** | 1.119 | **6.319*** | 2.497 | | 1.908 |
| | Retired | 78.53 (15.49) | 84.10 (10.81) | 55.90 (14.04) | 74.57 (15.29) | 21 (17.2) | 47.62 (19.21) | 65.57 (26.57) |
| | Unemployed | 78.30 (14.18) | 76.29 (16.43) | 58.02 (17.15) | 70.59 (16.05) | 62 (50.8) | 44.35 (18.34) | 66.62 (29.09) |
| | Employed | 82.99 (14.78) | 78.85 (14.73) | 55.75 (16.18) | 76.74 (14.76) | 39 (32.0) | 41.67 (20.94) | 72.38 (23.65) |
| Current caregiver | | -0.376 | -0.832 | -0.621 | -0.109 | -0.088 | | -0.020 |
| | Partner | 80.02 (14.93) | 77.91 (15.09) | 57.35 (15.27) | 73.26 (15.55) | 65 (53.3) | 44.23 (20.14) | 68.66 (26.40) |
| | Others | 79.49 (14.55) | 79.75 (14.39) | 56.52 (17.22) | 73.16 (15.90) | 57 (46.7) | 43.86 (18.47) | 67.65 (27.85) |
| Monthly household income (Yuan) | | **9.193*** | 3.947 | 0.324 | **14.334**** | 0.914 | | **8.267*** |
| | <3000 | 74.72 (13.54) | 75.22 (18.65) | 57.11 (15.94) | 65.28 (16.58) | 21 (17.2) | 41.67 (18.26) | 54.29 (35.99) |
| | 3000-5000 | 80.46 (14.53) | 77.92 (14.67) | 55.95 (15.72) | 74.01 (14.06) | 54 (44.3) | 45.37 (18.86) | 73.47 (21.74) |
| | >5000 | 82.10 (15.05) | 82.25 (11.05) | 57.96 (17.21) | 77.29 (15.33) | 47 (38.5) | 43.62 (20.50) | 70.62 (23.40) |
| Residence | | **-2.491*** | -1.286 | -0.401 | **-2.406*** | **-1.637** | | **-1.675** |
| | Urban area | 81.27 (14.85) | 80.04 (13.55) | 57.11 (16.71) | 75.09 (15.02) | 86 (70.5) | 45.64 (19.25) | 71.00 (24.68) |
| | Rural area | 76.74 (14.03) | 76.49 (16.70) | 56.57 (15.43) | 69.47 (16.43) | 36 (29.5) | 40.28 (19.16) | 62.47 (30.74) |
| Primary cancer | | 6.469 | 0.017 | 3.261 | 5.080 | 0.595 | | 0.144 |
| | Cervical | 78.01 (15.28) | 78.55 (15.61) | 55.90 (16.86) | 71.08 (17.16) | 47 (38.5) | 44.68 (19.46) | 66.61 (30.12) |
| | Ovarian/fallopian tube | 83.52 (14.10) | 79.33 (13.59) | 57.22 (13.88) | 75.65 (14.68) | 30 (24.6) | 42.50 (14.90) | 68.89 (24.61) |
| | Endometrial | 80.41 (14.77) | 79.21 (13.75) | 59.12 (17.22) | 75.58 (14.31) | 38 (31.1) | 45.39 (22.40) | 68.42 (26.83) |
| | Other | 75.56 (11.12) | 77.67 (17.82) | 53.33 (16.11) | 68.61 (13.72) | 7 (5.7) | 39.29 (19.67) | 73.33 (16.96) |
| Cancer stage | | 2.571 | 0.524 | 1.747 | 3.068 | 1.223 | | 5.552 |
| | I | 79.56 (13.76) | 79.70 (13.39) | 57.50 (15.97) | 74.08 (15.47) | 72 (59.0) | 44.79 (20.95) | 70.46 (25.93) |
| | II | 76.98 (19.35) | 76.67 (20.45) | 56.43 (18.45) | 68.06 (18.56) | 14 (11.5) | 42.86 (15.28) | 63.27 (31.81) |
| | III | 83.17 (14.07) | 81.73 (13.78) | 55.00 (14.28) | 74.68 (15.41) | 15 (12.3) | 43.33 (14.84) | 64.84 (23.98) |
| | IV | 79.52 (15.31) | 75.14 (15.55) | 56.71 (17.74) | 72.26 (14.82) | 21 (17.2) | 42.86 (19.59) | 65.71 (30.25) |

Note:

[†]Data were presented as Mean (SD); Mann-Whitney U tests were used for two independent samples; Kruskal-Wallis H tests were used for more than two independent samples; the correlation between continuous data was calculated using the Spearman correlation coefficient.

‡Relationship with God was answered only by the participants who responded to this item (n = 122).

*indicates $P < 0.05$,

**indicates $P < 0.001$.

**Table 5. Relationship between patients' characteristics and EORTC QLQ-SWB32 scales using multi-linear regression analysis.**

| Variables | Reference variables | B[†] | 95% CI[†] | Sig. | R | R² | Adjusted R² |
|---|---|---|---|---|---|---|---|
| **Dependent variables: RO** | | | | | 0.270 | 0.073 | 0.049 |
| Education-senior | Junior and below | 4.910 | −0.824, 10.645 | 0.093 | | | |
| Education-college and above | | 6.860 | 1.286, 12.435 | **0.016** | | | |
| Income < 3000 | Income > 5000 | −3.391 | −9.081, 3.020 | 0.298 | | | |
| Income 3000–5000 | | 0.773 | −4.111, 5.656 | 0.755 | | | |
| Rural residence | Urban | −1.259 | −6.137, 3.619 | 0.611 | | | |
| (Constant) | | 79.570 | 72.071, 87.068 | <0.001 | | | |
| **Dependent variables: RS** | | | | | 0.235 | 0.055 | 0.041 |
| Age | (Continuous) | 0.181 | −0.031, 0.392 | 0.093 | | | |
| Work-retired | unemployed | 6.036 | 0.282, 11.790 | **0.040** | | | |
| Work-employed | | 5.594 | 0.329, 10.859 | **0.037** | | | |
| (Constant) | | 66.927 | 55.623, 78.232 | <0.001 | | | |
| **Dependent variable: EX** | | | | | 0.364 | 0.132 | 0.101 |
| Education-senior | Junior and below | 6.970 | 0.785, 13.155 | **0.027** | | | |
| Education-college and above | | 8.668 | 2.222, 15.114 | **0.009** | | | |
| Income < 3000 | Income > 5000 | −7.807 | −14.350, −1.085 | **0.023** | | | |
| Income 3000–5000 | | −0.430 | −5.546, 4.685 | 0.868 | | | |
| Rural residence | Urban | −0.004 | −5.199, 5.191 | 0.999 | | | |
| Work-retired | unemployed | 0.542 | −5.362, 6.447 | 0.856 | | | |
| Work-employed | | −0.212 | −5.901, 5.476 | 0.941 | | | |
| (Constant) | | 72.069 | 63.690, 80.448 | <0.001 | | | |
| **Dependent variable: RG** | | | | | 0.437 | 0.191 | 0.156 |
| Age | (Continuous) | 0.114 | −0.198, 0.426 | 0.469 | | | |
| Religious belief | No | 30.315 | 15.118, 45.511 | **<0.001** | | | |
| Education-senior | Junior and below | −10.969 | −20.135, −1.802 | **0.019** | | | |
| Education-college and above | | −3.332 | −12.693, 6.028 | 0.482 | | | |
| Rural residence | Urban | −10.221 | −17.878, −2.564 | **0.009** | | | |
| (Constant) | | 52.622 | 29.973, 75.270 | <0.001 | | | |

Note:

[†] Unstandardized coefficients beta with 95% CI for B;

[‡] Model summary: RO, F = 3.063, $P = 0.011$; RS, F = 3.811, $P = 0.011$; EX, F = 4.178, $P < 0.001$; RG, F = 5.465, $P < 0.001$;

[§] Education background, work status, and family income per month were converted to three dummy variables;

[¶] Forced entry method was used in the muti-linear regression analysis;

cancer types, and reported lower scores across most dimensions than our study found, except for Existential.

Sun used principal component analysis, and identified four dimensions for the SWB32. The items comprising each dimension identified did not precisely match the original English version, and Sun therefore modified the items included in each scale, including expanding the RO scale from six to seven items and reducing the RS scale from five to four items. Discrepancies therefore exist between outcomes from our current study, the main validation study [17], and Sun's study [29,39]. In contrast, Feng et al. [28] used the original scale's scoring methodology, yielding results similar to our present study's findings.

There are some distinctions in content between our authorized translation and the preceding one. These variances are evident in the interpretation of certain items. For example, item 13, 'valued as a person,' was translated as '一个有价值的人 (back translation in English: a

valuable person) in the unapproved version. However, according to the EORTC Translation Unit, the intended meaning should convey that the patient feels respected by others or thinks that they are important and special;' the respondent's view of how other people regard them. We therefore translated it as '作为人被重视,' aligning with this explanation, so our approved translation is closer to the precise meaning intended in the English SWB32.

The mean scores for the main EORTC QLQ-SWB32 scales: RO, RS, RSG, EX, and also for Global-SWB in our study were similar to previous studies using the version in other languages [40–42], indicating that our approved Chinese translation equates well to what the scales in the original version seek to measure. Our 122 study participants' mean RG scores were the lowest (44.06) of all their SWB32 mean scale scores. Previous studies in other countries have reported higher mean RG scores: 74.9 in Cyprus [41], and 67.7 in Finland [42]. Notably, neither of the two earlier studies in China [28,29,39] reported RG scale scores.

RG is only answered by those with faith, so RG scores in different countries cannot be simply, directly compared. Religion, and religious faith and belief, also mean different things in different countries and cultures. China is generally less religiously oriented than many other countries, but Chinese people may also be different from people from cultures with no religion at all, because of the influence of aspects of Chinese traditional culture such as Confucianism, Taoism, and Buddhism [28]. This particular cultural background provides a religion-like spiritual function [22]. Many Chinese people identify themselves as nonbelievers but still hold religious-like beliefs and behave in ways that can be similar to religious behaviors [18]. However, these cultural influences do not correspond to the concept of 'God' as understood in monotheistic religions prevalent in Western cultures. The concept of 'God' in Chinese culture tends to be more abstract, often associated with ideas such as heaven, fate, or ancestral spirits, rather than a personal deity [23,43]. Therefore, when asked about a 'Relationship with God,' Chinese participants may interpret the question differently, leading to lower scores. In fact, this reflects cultural differences in spirituality rather than a lack of spiritual wellbeing. This cultural background may result in different interpretations and response patterns to RG items. Although 96.5% of our participants identified as non-religious, contrasting sharply with the international validation sample (34.6% non-religious; 41.7% Christian, 11.1% Muslim) [17], 61% of our respondents answered the RG single item, similarly to the international validation study of SWB32 finding that over a half self-declared non-religious individuals still responded to the skip items on the SWB measure [17]. However, those patients who had identified themselves as religious scored higher on RG than those who had not. A previous study, also with Chinese gynecological cancer patients, found that explicit religious beliefs correlated with spiritual well-being [44]. This may be related to the belief that God, or a greater power will give strength and help inner peace.

Our study also identified several other socio-demographic variables that could impact spiritual well-being in gynecological cancer patients. Patients with a higher level of education scored higher on RO and EX, similar to the findings of Wang et al. [45], who investigated spirituality in Chinese stroke patients. More educated patients may be better at maintaining physical, spiritual, and social wellbeing, and adept at utilizing all available resources for positive psychological suggestions and adjustments to maintain overall spiritual wellbeing and peace of mind[38]. As for employment and economic status, we found that patients who were unemployed or had lower incomes had lower scores on RS and EX, respectively. The results were consistent with a previous study by Dabo et.al [40] in Croatian cancer patients. Unemployed patients may have low incomes. During anti-cancer treatment, these patients may be constrained and limited by their financial difficulties, face related stresses and other difficulties [40]. We found no significant differences for cancer

stage for our participants' scores, similar results were found in previous studies that cancer stage was not associated with spiritual wellbeing[46,47]. Even in the early stages of the disease, healthcare professionals should assess the spiritual wellbeing of cancer patients and provide appropriate spiritual care.

## 4.1  Relevance for clinical practice

Spiritual well-being and the provision of spiritual care are integral components of cancer care. Our validation study has contributed to confirming that the EORTC QLQ-SWB32 is a valid and reliable measurement of spiritual wellbeing in Chinese cancer patients. This measure could help identify cancer patients with lower spiritual wellbeing and unmet spiritual needs. Healthcare professionals can also use this tool to initiate discussions surrounding spiritual concerns and to evaluate responses to spiritual care interventions in China.

It is important to note that all our participants were gynecological patients, so female, and more than half with stage I cancer, whereas other studies have involved patients of mixed sexes, and with a variety of cancers and cancer stages. Rohde et al. [15] found differences related to the sex of the main validation study participants. Nevertheless, the demographic characteristics identified in our study of this group which were associated with variations in spiritual wellbeing can serve as valuable indicators for healthcare providers. These findings underscore the potential benefits of intensified spiritual care for patients with lower educational attainment, individuals who are unemployed or with lower incomes, and those without religious beliefs. Strategies for enhancing spiritual wellbeing include meaningful/existential interventions [48] and psychosocial interventions such as creative arts and yoga [4]. Additionally, integrating the EORTC QLQ-SWB32 into routine monitoring of treatment and care warrants consideration. Healthcare professionals must fortify their spiritual care competencies, acquiring profound knowledge and expertise in assessing patients' spiritual concerns and the skills to address them effectively. This endeavor will comprehensively enhance the quality of spiritual care delivery within cancer care [49].

Given the absence of recommended cut-off scores by the original validated literature and a definitive 'gold standard' in this domain, establishing which scale scores denote high spiritual wellbeing remains to be discovered. We therefore call for a broader adoption of the EORTC QLQ-SWB32 by researchers, so as to explore this and other questions further, including sex differences.

## 4.2  Study limitations

Our study had some limitations. As noted, the absence of established cut-off scores for the EORTC QLQ-SWB32 poses a challenge in categorizing scores as indicative of high or low levels of spiritual wellbeing. Our study is constrained to score comparisons with existing studies identifying scale score minima within the measure. Furthermore, our regression analysis yielded a limited interpretation of the impact of demographic characteristics on spiritual wellbeing. This suggests that our study might have recorded other variables, encompassing additional demographic, clinical, and psychosocial dimensions, which may impact spiritual wellbeing. Our cross-sectional design also limited the possibility of identifying any causal associations between the variables we included. In contrast to the main international validation study [14], and the studies in Cyprus [35] and Finland [36], but similar to the Croatian study [34], more than half of our study participants had a stage I cancer, which prevented comparisons of scores by cancer stage. Last but not least, our validation study was conducted

only with gynecological cancer patients, i.e., women only. Whether the SWB32 is suitable for male patients and other cancer patients in China requires future research. One additional limitation of this study is the absence of effect size calculations. While statistical significance was assessed, effect size measures were not reported due to software constraints and the need for additional manual calculations. Future studies should include effect size to provide a more comprehensive understanding of the practical significance of the findings.

## 5  Conclusions

In conclusion, our translation of the EORTC QLQ-SWB32 into Chinese demonstrated high reliability and validity among gynecological cancer patients. The dimensions of this Chinese version are consistent with the original tool. Higher spiritual wellbeing was associated with higher education levels, higher incomes, work or retirement, and religious beliefs. Conversely, patients with lower education levels, unemployed or with lower incomes, and those who said they had no religious faith scored lower on several of the SWB32 scales. Healthcare providers should consider these factors when providing spiritual care to cancer patients.

The EORTC QLQ-SWB32 was developed to facilitate the measurement of spiritual wellbeing in multiple linguistic and cultural contexts, and enable comparisons between them, so it is undoubtedly beneficial for different studies worldwide to use the same measure. Nevertheless, results from studies in different countries and cultural contexts need to be examined and compared with caution because various aspects of spirituality may be present in every culture, although they may not be explicitly defined as such. An in-depth understanding of traditions and social networks and connections is needed to fully contextualise and understand results from studies in this field.

## Supporting information

**S1 Appendix.  Data.**
(CSV)

**S2 Appendix.  Specimen SWB32 Chinese Mandarin (China) version.**
(PDF)

**S3 Appendix.  Specimen SWB32 English version.**
(PDF)

**S4 Appendix.  Global Research Questionnaire.**
(DOCX)

## Acknowledgments

The authors thank all the medical professionals involved in our study for their help and support in this study. Thanks to all the patients who participated in this study.

## Author contributions

**Conceptualization:** Jing Chen, Bella Vivat.

**Formal analysis:** Yue Feng, Xiujing Guo.

**Investigation:** Tangwei Lin, Xingcan Liu.

**Methodology:** Yue Feng, Jiangshan Luo, Xiujing Guo, Jing Chen.

**Writing – original draft:** Yue Feng, Jing Chen.

**Writing – review & editing:** Bella Vivat.

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
