## [Decision Letter · Decision Letter 0]

3 Jan 2025

PONE-D-24-36540Translation and validation of the Chinese version of EORTC QLQ-SWB32 assessing the spiritual wellbeing of women with gynecological cancerPLOS ONE

Dear Dr. Chen,

Thank you for submitting your manuscript to PLOS ONE. After careful consideration, we feel that it has merit but does not fully meet PLOS ONE’s publication criteria as it currently stands. Therefore, we invite you to submit a revised version of the manuscript that addresses the points raised during the review process.

We look forward to receiving your revised manuscript.

Kind regards,

Mirosława Püsküllüoğlu, MD, PhD

Academic Editor

PLOS ONE

Journal Requirements:

Reviewers' comments:

Reviewer's Responses to Questions

**Comments to the Author**

1. Is the manuscript technically sound, and do the data support the conclusions?

Reviewer #1: Yes

Reviewer #2: Yes

Reviewer #3: Yes

2. Has the statistical analysis been performed appropriately and rigorously? 

Reviewer #1: I Don't Know

Reviewer #2: Yes

Reviewer #3: Yes

3. Have the authors made all data underlying the findings in their manuscript fully available?

Reviewer #1: Yes

Reviewer #2: Yes

Reviewer #3: Yes

4. Is the manuscript presented in an intelligible fashion and written in standard English?

Reviewer #1: Yes

Reviewer #2: Yes

Reviewer #3: Yes

5. Review Comments to the Author

Reviewer #1: The manuscript describes the translation and validation of the EORTC QLQ-SWB32 instrument for assessing spiritual well-being among Chinese gynecological cancer patients. The research methodology and statistical analyses are robust, and the findings contribute to the field of cross-cultural validation of quality-of-life measures.

1. The study is original, focusing on translating and validating the EORTC QLQ-SWB32 for a specific cultural and linguistic context. This is a novel contribution as it expands the utility of the measure for Chinese-speaking populations.

2. There is no indication that the results have been published elsewhere.

3. The methodology, including translation steps, pilot testing, and validation phases, is described in detail and follows the EORTC guidelines. However, given the complexity of the methods, an expert reviewer with in-depth knowledge of psychometric validation and advanced statistical techniques should assess the rigor and appropriateness of these methods to ensure their accuracy and alignment with best practices.

4. The conclusions are supported by the data. However, the discussion could benefit from:

4.1 addressing potential cultural nuances influencing the responses, particularly regarding the "Relationship with God" dimension, given the reported low scores

4.2 Delving deeper into cultural interpretations of spirituality in Chinese contexts and how they may affect responses to specific items in the SWB32 scale.

4.3 Comparing findings more extensively with international validations of the SWB32 to contextualize the results.

5. The manuscript is written in standard academic English. Figures and tables are clear. However:

5.1 there are minor grammatical errors and inconsistencies in terminology (e.g., switching between "participants" and "patients").

5.2 the exploratory graph analysis figure could benefit from a more detailed legend explaining the clusters and correlations

6. The study adheres to ethical standards, with approval from the appropriate ethics committee and written informed consent obtained from participants.

7. The manuscript adheres to reporting standards for validation studies, providing sufficient detail on the translation and validation process. Data availability is clearly stated.

The manuscript addresses an important gap in the literature. However, revisions are necessary to improve clarity, cultural contextualization, and the discussion of findings. I recommend acceptance after minor revisions.

Reviewer #2: The aim of this study is the translation and validation of the Chinese version of EORTC QLQ-SWB32 assessing the spiritual wellbeing of women with gynecological cancer.

This article is well-structured and well-written. In the introduction, it provides a complete overview of the situation related to ovarian cancer in China, with a mention of improving the quality of life in patients with this type of cancer, the questionnaire related to spiritual wellbeing, and religious aspects in China.

The statistical analysis is well-structured and compliant. The use of effect size is suggested, as it would improve the interpretation of the results (especially in the regression models). Additionally, all software used should be specified in the methods section for clarity. The use of different analysis software makes the reproducibility of the study complex.

Detailed comment #1

From line 141 to 148: Reformulate di paragraph to better clarify the decision to use the non parametric test.

Please include where possible effect size.

Reviewer #3: This is a straightforward evaluation of a Chinese language version of the European Organization for Research and Treatment of Cancer measure of spiritual well-being. The authors followed the translation guidelines for establishing content validity, reliability, and construct validity.

Overall, the validation study has been competently conducted. There are a few minor comments that may help strengthen the manuscript.

1. Line 145. Please specify which “Nonparametric test” correlation was used.

2. Please provide more information on the questionnaire used to measure the demographics, especially the units of measure for certain variables, especially income.

3. Table 3. It would help to rearrange the ordering of the factors across the table to be consistent with the order in which they are presented in Table 2.

4. Table 4 is difficult to follow, probably due to the formatting. For example, terms inside parentheses wrap around, and the items in the right-hand columns do not appear aligned. Please ensure that the table is in a readable format.

6. PLOS authors have the option to publish the peer review history of their article (what does this mean? ). If published, this will include your full peer review and any attached files.

**Do you want your identity to be public for this peer review?** For information about this choice, including consent withdrawal, please see our Privacy Policy .

Reviewer #1: No

Reviewer #2: No

Reviewer #3: No

---

## [Author Response · Author response to Decision Letter 1]

17 Feb 2025

Ref: Manuscript ID PONE-D-24-36540

Translation and validation of the Chinese version of EORTC QLQ-SWB32 assessing the spiritual wellbeing of women with gynecological cancer

PLOS ONE

Dear Dr Mirosława Püsküllüoğlu and Reviewers,

On behalf of my co-authors, we thank you for reviewing this manuscript and allowing us to revise it. Please find attached the revised version of our manuscript, "Translation and validation of the Chinese version of EORTC QLQ-SWB32 assessing the spiritual wellbeing of women with gynecological cancer."

The reviewers' comments were highly insightful and enabled us to improve the quality of our manuscript significantly. I appreciate all your comments and suggestions. The following pages contain our point-by-point responses to each reviewer's comments.

Following the Reviewer's suggestions and comments, we have carefully revised the manuscript. In the manuscript file, we highlighted the parts in yellow that need correction and supplementation. In the response letter, our responses are in blue, and quoted texts from the manuscript are in green and italics. We hope the revisions and accompanying responses will be sufficient to make our manuscript suitable for publication in PLOS ONE.

Special thanks for the precious time you spent making constructive remarks. We look forward to hearing from you at your earliest convenience.

Best regards,

Jing Chen

West China Second University Hospital,

Sichuan University

Email: chenjing_scu@sina.com

Reviewer #1:

• The methodology, including translation steps, pilot testing, and validation phases, is described in detail and follows the EORTC guidelines. However, given the complexity of the methods, an expert reviewer with in-depth knowledge of psychometric validation and advanced statistical techniques should assess the rigor and appropriateness of these methods to ensure their accuracy and alignment with best practices.

Response:

Thank you for your insightful comment. The translation process of SWB32 strictly followed the EORTC translation procedure. Our research team conducted two forward translations, an initial linguistic reconciliation, and two backward translations of the reconciled version. The EORTC Translation Unit carefully reviewed each step of this process, and we engaged in multiple rounds of discussion and revisions with them. These ensured that the final Chinese version accurately reflected the original scale's intended meaning before we proceeded with pilot testing.

Given this rigorous translation process and the direct involvement of the EORTC Translation Unit- which specializes in ensuring linguistic and conceptual equivalence across languages, we did not conduct an additional review by independent experts in psychometric validation and advanced statistical techniques during this phase. Instead, our approach prioritized the established best practices outlined by the EORTC guidelines to maintain cross-cultural validity.

We acknowledge that the lack of an additional expert reviewer for psychometric validation may be a limitation of our study. However, our adherence to EORTC's well-validated methodology provides a robust foundation for linguistic and conceptual accuracy.

• Addressing potential cultural nuances influencing the responses, particularly regarding the "Relationship with God" dimension, given the reported low scores.

Response:

We appreciate your insightful comment regarding cultural nuances in interpreting the "Relationship with God" (RG) dimension. We fully agree that cultural context significantly affects how participants understand and respond to RG items, especially within the Chinese cultural background.

In Chinese cultures, spirituality is often articulated in ways that differ from Western religious frameworks; it frequently emphasizes concepts like harmony, ancestral reverence, and philosophical traditions rather than a direct and personal relationship with a deity. These distinct approaches to spiritual expression may affect how participants interpret and respond to items within this dimension. We recognize this as an essential factor and have discussed how cultural nuances may influence responses.

Quoted texts:

Line 343: However, these cultural influences do not correspond to the concept of 'God' as understood in monotheistic religions prevalent in Western cultures. The concept of 'God' in Chinese culture tends to be more abstract, often associated with ideas such as heaven, fate, or ancestral spirits, rather than a personal deity [1,2]. Therefore, when asked about a 'Relationship with God,' Chinese participants may interpret the question differently, leading to lower scores. In fact, this reflects cultural differences in spirituality rather than a lack of spiritual wellbeing. This cultural background may result in different interpretations and response patterns to RG items.

• Delving deeper into cultural interpretations of spirituality in Chinese contexts and how they may affect responses to specific items in the SWB32 scale.

Response:

Thanks for your valuable feedback. We have carefully considered your suggestions and have made the necessary revisions. In response to your comment, we have expanded on the cultural context of spirituality, specifically focusing on how traditional Chinese cultural frameworks (such as Confucianism, Taoism, Buddhism, and folk beliefs) shape spiritual wellbeing. Besides, we think that part of the revisions of the last comment can also help respond to this comment.

Quoted texts:

Introduction section

Line 81: Chinese culture traditionally emphasizes harmony, collectivism, and moral ethics, which significantly influence spiritual perceptions[2]. Unlike Western conceptions of spirituality that often center on personal relationships with a transcendent deity, Chinese spirituality is deeply rooted in the concepts of interconnectedness and balance[2,3]. This is reflected in Confucianism's emphasis on social harmony, Taoism's pursuit of unity with nature, and Buddhism's focus on inner enlightenment. These philosophical traditions contribute to a holistic worldview where spirituality is integrated into everyday life rather than confined to organized religious practices[4]. Consequently, spirituality in Chinese culture is more likely to be expressed through ethical living, social roles, and relational harmony rather than explicit religious beliefs [4].

Line 91: Researchers have recognized the importance of spirituality within Chinese contexts, defining it as a relationship with self, others, nature, and Higher Being(s) [5,6]. These dimensions align with the four key scoring areas of SWB32.

Discussion section

Line 343: However, these cultural influences do not correspond to the concept of 'God' as understood in monotheistic religions prevalent in Western cultures. The concept of 'God' in Chinese culture tends to be more abstract, often associated with ideas such as heaven, fate, or ancestral spirits, rather than a personal deity [1,2]. Therefore, when asked about a 'Relationship with God,' Chinese participants may interpret the question differently, leading to lower scores. In fact, this reflects cultural differences in spirituality rather than a lack of spiritual wellbeing. This cultural background may result in different interpretations and response patterns to RG items.

• Comparing findings more extensively with international validations of the SWB32 to contextualize the results.

Response:

We appreciate your suggestion to contextualize our findings with the international validation of the EORTC SWB32. We have expanded our discussion to compare our results with the original international validation study. This includes the comparison of the structure of the SWB32, the cultural nuances, and the religious beliefs of the participants.

Quoted texts:

Line 294: The internal consistency and split-half reliability were generally strong for four factors, and the results were similar to the internationally validated version [7]. The four-factor structure identified from EFA, CFA, and EGA in our study aligns closely with the international study's principal component and Rasch-derived scales: Relationships with Others (RO), Relationship with Self (RS), Relationship with Someone or Something Greater (RSG), and Existential (EX).

……

Line 350: Although 96.5% of our participants identified as non-religious, contrasting sharply with the international validation sample (34.6% non-religious; 41.7% Christian, 11.1% Muslim) [7], 61% of our respondents answered the RG single item, similarly to the international validation study of SWB32 finding that over a half self-declared non-religious individuals still responded to the skip items on the SWB measure [7].

• There are minor grammatical errors and inconsistencies in terminology (e.g., switching between "participants" and "patients").

Response:

Thank you for pointing out the grammatical errors and inconsistencies in terminology. Throughout the text, we have standardized the wording by consistently referring to the study population as patients. Since the study focuses on gynecological cancer patients and their spiritual wellbeing in a medical context, it is more appropriate to use 'patients' instead of 'participants'.

• The exploratory graph analysis figure could benefit from a more detailed legend explaining the clusters and correlations.

Response:

Thanks for your valuable feedback. We have made some modifications about the figure legend according to your suggestions.

Quoted texts:

Line 240: Fig 1. Exploratory graph analysis of the four factors of SWB32.

† Colored cluster: Red (group 1), Relationships with Others; Blue (group 2), Relationship with Self; Green (group 3), Relationship with Someone or Something Greater; Orange (Group 4), Existential.

‡ Nodes (circles) represent the items in SWB32. Each node corresponds to a specific item measuring different aspects of spiritual wellbeing. Edges (lines) represent partial correlations between items. The thickness of the edges represents the magnitude of the correlation; the thicker the edge, the stronger the correlation. Green edges= positive correlations, red edges= negative correlations.

§ The RG scale was answered only by the patients who responded 2-4 on items 21 and 22 (n=122).

Reviewer #2

From line 141 to 148: Reformulate di paragraph to better clarify the decision to use the nonparametric test. Please include where possible effect size.

Response:

Thanks for your valuable feedback. We have revised the paragraph in the Statistical section. Given the skewed distribution of SWB32 scores, we opted for nonparametric methods to ensure the robustness of our analysis. Specifically, we used the Mann-Whitney U test or Kruskal- Wallis H test, where applicable variables. We acknowledge that we didn't report effect size measures such as rank-biserial correlation or Cliff's Delta for the Mann-Whitney U test. The primary reason for this omission is that SPSS does not automatically calculate these effects sizeds for nonparametric tests, and additional manual computation or external software would be required. Given the scope of our study, our primary focus was on statistical significance rather than effect size estimation.

However, we recognize that effect size provides valuable information regarding the practical signicance of our findings. We acknowledge this limitation and will consider incorporating effect size measures in future research to enhance the interpretability of our results. We have added this point to the Limitations section.

Quoted texts:

Statistical analysis

Line 158: We assessed relationships between spiritual wellbeing and countable variables using the Nonparametric Test. Mann-Whitney U tests were used for comparisons between two groups; Kruskal- Wallis H tests were used for more than two groups.

Limitation

Line 416: One additional limitation of this study is the absence of effect size calculations. While statistical significance was assessed, effect size measures were not reported due to software constraints and the need for additional manual calculations. Future studies should include effect size to provide a more comprehensive understanding of the practical significance of the findings.

Reviewer #3

1. Line 145. Please specify which "Nonparametric test" correlation was used.

Response:

Thanks for your comment. We have revised the paragraph in the Statistical section. Given the skewed distribution of SWB32 scores, we opted for nonparametric methods to ensure the robustness of our analysis. Specifically, we used the Mann-Whitney U test or Kruskal- Wallis H test, where applicable.

Quoted texts:

Statistical analysis

Line 158: We assessed relationships between spiritual wellbeing and countable variables using the Nonparametric Test. Mann-Whitney U tests were used for comparisons between two groups; Kruskal- Wallis H tests were used for more than two groups.

2. Please provide more information on the questionnaire used to measure the demographics, especially the units of measure for certain variables, especially income.

Response:

Thanks for your valuable feedback. We have revised our manuscript to clarify the demographic variables and their units of measurement. Specifically, age is recorded in years, and monthly family income is reported in Yuan. Additionally, we collected information on ethnic group, marital status, working status, educational background, and religious beliefs or practices. Clinical data, including diagnosis and cancer stage, were obtained from patients' medical records.

Quoted texts:

Line 124: We collected relevant demographic information, including age (years), ethnicity, marital status, employment status, educational background, and monthly household income (Yuan). Clinical data, such as diagnosis and cancer stage, were extracted from patients' medical records. Additionally, we inquired whether patients held religious beliefs or actively engaged in religious practices.

3. Table 3. It would help to rearrange the ordering of the factors across the table to be consistent with the order in which they are presented in Table 2.

Response:

Thank you for your helpful suggestion. We have revised the ordering of factors in Table 3 to ensure consistency with Table 2.

4. Table 4 is difficult to follow, probably due to the formatting. For example, terms inside parentheses wrap around, and the items in the right-hand columns do not appear aligned. Please ensure that the table is in a readable format.

Response:

Thank you for your feedback. As per the journal's formatting requirements, tables must be inserted immediately after their first mention in the manuscript. Table 4 contains wide content, which may be better displayed in a landscape layout. This may affect readability in the current portrait format. To assist in reviewing the table, we have attached a landscape-format screenshot in our response letter. We appreciate your understanding.

Reference

1. Saroglou V, Clobert M, Cohen AB, Johnson KA, Ladd KL, Van Pachterbeke M, et al. Believing, Bonding, Behaving, and Belonging: The Cognitive, Emotional, Moral, and Social Dimensions of Religiousness across Cultures. J Cross Cult Psychol. 2020;51:551–75.

2. Tian Q, Becker M, Hilton D, Tian Q, Becker M, Hilton D. Mind Attribution to Gods and Christians in the Chinese Cultural Context. Scientific Study of Religion. 2023;62:885–900.

3. Zhang C, Lu Y, Sheng H. Exploring Chinese folk religion: Popularity, diffuseness, and diversities. Chinese Journal of Sociology. 2021;7:575–92.

4. Luo W, Chen F, Luo W, Chen F. The Salience of Religion Under an Atheist State: Implications for Subjective Well-Being in Contemporary China. Soc Forces. 2021;100:852–78.

5. Niu Y, McSherry W, Partridge M. Exploring the meaning of spirituality and spiritual care in Chinese contexts: A scoping review. Journal of Religion and Health. 2022;61:2643–62.

6. Chao C-SC, Chen C-H, Yen M. The essence of spirituality of terminally ill patients. The journal of nursing research : JNR. 2002;10:237–45.

7. Vivat B, Young T e., Winstanley J, Arraras J i., Black K, Boyle F, et al. The international phase 4 validation study of the EORTC QLQ-SWB32: A stand-alone measure of spiritual well-being for people receiving palliative care

---

## [Decision Letter · Decision Letter 1]

11 Mar 2025

Translation and validation of the Chinese version of EORTC QLQ-SWB32 assessing the spiritual wellbeing of women with gynecological cancer

PONE-D-24-36540R1

Dear Dr. Chen,

We’re pleased to inform you that your manuscript has been judged scientifically suitable for publication and will be formally accepted for publication once it meets all outstanding technical requirements.

Kind regards,

Boshra A. Arnout

Academic Editor

PLOS ONE

Additional Editor Comments (optional):

Reviewers' comments:

Reviewer's Responses to Questions

**Comments to the Author**

1. If the authors have adequately addressed your comments raised in a previous round of review and you feel that this manuscript is now acceptable for publication, you may indicate that here to bypass the “Comments to the Author” section, enter your conflict of interest statement in the “Confidential to Editor” section, and submit your "Accept" recommendation.

Reviewer #1: All comments have been addressed

Reviewer #3: All comments have been addressed

2. Is the manuscript technically sound, and do the data support the conclusions?

Reviewer #1: Yes

Reviewer #3: Yes

3. Has the statistical analysis been performed appropriately and rigorously? 

Reviewer #1: Yes

Reviewer #3: Yes

4. Have the authors made all data underlying the findings in their manuscript fully available?

Reviewer #1: Yes

Reviewer #3: Yes

5. Is the manuscript presented in an intelligible fashion and written in standard English?

Reviewer #1: Yes

Reviewer #3: Yes

6. Review Comments to the Author

Reviewer #1: Manuscript PONE-D-24-36540R1

I appreciate the authors’ thorough responses to my comments and their careful revisions of the manuscript. They have addressed all concerns appropriately, improving the clarity, methodological rigor, and overall quality of the study. Given these revisions, I now find the manuscript suitable for publication.

I recommend the manuscript for acceptance in its current form.

Best regards!

Reviewer #3: Minor editorial comment: Line 37. I think “by men” should read “than men”

It is unfortunate that the authors did not provide effect sizes. Their response to the reviewer's comments is not convincing.

7. PLOS authors have the option to publish the peer review history of their article (what does this mean? ). If published, this will include your full peer review and any attached files.

**Do you want your identity to be public for this peer review?** For information about this choice, including consent withdrawal, please see our Privacy Policy .

Reviewer #1: No

Reviewer #3: No

---

## [Editor Report · Acceptance letter]

PONE-D-24-36540R1

PLOS ONE

Dear Dr. Chen,

I'm pleased to inform you that your manuscript has been deemed suitable for publication in PLOS ONE. Congratulations! Your manuscript is now being handed over to our production team.

Kind regards,

on behalf of

Professor Boshra A. Arnout

Academic Editor

PLOS ONE